# Single Lead Epidural Spinal Cord Stimulation Targeted Trunk Control and Standing in Complete Paraplegia

**DOI:** 10.3390/jcm11175120

**Published:** 2022-08-30

**Authors:** Ashraf S. Gorgey, Jan J. Gouda

**Affiliations:** 1Spinal Cord Injury and Disorders Center, Hunter Holmes McGuire VAMC, 1201 Broad Rock Boulevard, Richmond, VA 23249, USA; 2Department of Physical Medicine & Rehabilitation, Virginia Commonwealth University, Richmond, VA 23298, USA; 3Neurosurgery Department, Louran Hospital, Alexandria 5451110, Egypt; 4Department of Surgery, Wright State University, Dayton, OH 45435, USA

**Keywords:** spinal cord injury, epidural stimulation, percutaneous leads, migration, motor control

## Abstract

A 25-year-old male with T3 complete AIS A was implanted with percutaneous spinal cord epidural stimulation (scES; eight contacts each) leads and a Medtronic Prime advance internal pulse generator. The two leads were placed at the midline level to cover the region of the T11–T12 vertebrae. Five days after implantation, X-ray showed complete migration of the left lead outside the epidural space. Two weeks after implantation, reprogramming of the single right lead (20 Hz and 240 µs) after setting the cathode at 0 and the anode at 3 resulted in target activation of the abdominal muscles and allowed for the immediate restoration of trunk control during a seated position, even with upper extremity perturbation. This was followed by achieving immediate standing after setting the single lead at −3 for the cathode and +6 for the anode using stimulation configurations of 20 Hz and 240 µs. The results were confirmed with electromyography (EMG) of the rectus abdominus and lower extremity muscles. Targeted stimulation of the lumbosacral segment using a single lead with a midline approach immediately restored the trunk control and standing in a person with complete paraplegia.

## 1. Case Report

Spinal cord epidural stimulation (scES) has been historically used for pain management [1]. The scES involves stimulation of the ascending fibers in the dorsal column for several pain managing disorders similar to failed back pain surgery, recurrent back pain syndrome, and lower extremity radiated pain [1,2]. It has not been until recently that the applications of scES have emerged for the restoration of motor recovery in neurological disorders, especially in persons with spinal cord injury (SCI) [3,4,5]. We have previously used scES to enhance motor and autonomic functions in a person with complete SCI [6,7]. Unlike paddle implantation, percutaneous lead implantation provides access to the epidural space without the need to perform surgical laminectomy or laminotomy. This is likely to reduce the spinal biomechanical disadvantages associated with laminectomy [8,9]. Furthermore, percutaneous scES offers immediate post-operative recovery and discharge from the hospital on the same day following implantation.

Using a single percutaneous scES lead, mapping of the lumbosacral segments of the spinal cord was performed in 34 subjects with upper motor neuron neurological disorders [10]. The midline placement of percutaneous scES resulted in posterior root reflex stimulation of the target muscles based primarily on the location of the cathode [10]. Furthermore, percutaneous scES was previously used to reduce spasticity and enhance the voluntary motor control in persons with SCI [11,12]. We have recently shown that configuring a single lead to initiate rhythmic EMG activity enabled motor recovery in two people with motor complete SCI (Gorgey et al., unpublished data). Enhancement in motor recovery using a single lead occurred despite the migration of both percutaneous leads (Gorgey et al., unpublished data). The improvement in motor recovery was demonstrated in the form of enhanced exoskeletal performance, the initiation of knee extensor peak torque, trunk control, and overground locomotion (Gorgey et al., unpublished data). The aforementioned findings support the hypothesis that a single midline lead may be capable of enabling motor recovery. However, we are unaware of any studies that have examined the effect of a single scES percutaneous lead on motor recovery in people with complete SCI. Tonic sustained electrical activity may facilitate the restoration of functional independence in people with SCI. Benavides et al. showed that transspinal stimulation adjusted to deliver tonic stimulation resulted in increasing the cervical network excitability and improving the upper extremity functions in people with tetraplegia [13]. We hereby relied on inducing functional motor patterns (i.e., trunk control and standing) followed by gradually enabling motor control by actively engaging the supraspinal control of the desired posture [14].

Here, we present the case of the T3 complete SCI participant who underwent scES implantation. Five days after implantation, X-ray showed the complete migration of one of the leads outside the epidural space. We were capable of reprograming and configuring the right single lead to map the corresponding muscles and immediately restore trunk control and standing.

A 25-year old male experienced a road traffic accident approximately 3 years and 8 months before conducting surgical spinal fusion that extended from T2–T7 (Figure 1a). Multiple injuries were sustained including pulmonary contusions, abdominal visceral injuries, and pelvic fractures in addition to severe fracture dislocation of the fourth and fifth thoracic vertebrae. Prior to implantation, the participant underwent a detailed American Spinal Cord Injury Impairment Scale (AIS) classification and a neurological examination that indicated positive patellar tendon reflex and Babinski reflex (see Appendix A).

On 27 February 2022, the participant underwent implantation of scES in a specialized operating suite under fluoroscopy guidance at Louran Hospital in Alexandria, Egypt. Prior to implantation, the participant signed a consent form ensuring complete understanding of the procedure of scES implantation and this was conducted solely for research purposes. Clinically, the procedure was performed primarily for managing and controlling spasticity. A Medtronic Prime advance Internal pulse generator with two 90 cm leads and eight contacts each were used (Figure 1b). This length of the percutaneous leads may be possibly attributed to later migration in one of the leads.

Surgery lasted for one and a half hours with less than 10 mL of blood loss. The operation was uneventful with smooth post operative recovery. An intra-operative trial was performed to ensure the lower extremity muscle contraction without EMG recording. Intra-operative and immediate postoperative fluoroscopy showed optimal positioning of the leads spreading from the midline of the T11 to T12 vertebral levels (Figure 1b). The participant was then discharged from the hospital on the third day. Unfortunately, on 3 March, the anterior–posterior X-ray view showed migration of the left lead outside the epidural space, whereas the right lead migrated from the proximal rim of T11 to the upper 1/3 of the T12 vertebrae (Figure 2) [15]. The migration of the leads occurred during his travel home to Cairo, Egypt (~250 km).

Two weeks after the surgery, spinal motor mapping was performed using Sayenko’s approach, as listed in Figure 3. Before the mapping procedures, the participant resting blood pressure was 103/57 mmHg, a heart rate of 97 bpm, and modified Ashworth scale of grades 2 and 3 for the left and right legs, respectively. Prior to mapping, the participant underwent a full AIS exam to confirm his previous classification, document any zones of partial preservation, and ensure that no sacral sparing (i.e., clinically complete injury; see Appendix A). The goal of spinal mapping was to determine the best configuration that can elicit flexion or extension synergies of the lower extremity muscles. Based on Sayenko’s approach [16], we delivered five programs from A1–A5 in the supine lying position (Figure 3). To narrow down the selection among the five programs, stimulation pulses were initially delivered at a frequency of 2 Hz, pulse duration of 150 µs, and amplitude of the current that gradually increased from 0–10 volts (Table 1). The current was gradually increased until a visible response was noted and later confirmed with electromyography (EMG; Figure 4A–C). On the second day, the same program was repeated in the supine lying position to ensure the target achievements of functional movements at 20–30 Hz and 210 µs. The results and the progression of these programs delivered on two consecutive days are listed in Table 1. Our findings indicated that the A3 program is the most prominent program to achieve bilateral leg extension, which can be translated into the standing position (Table 1).

As a result of migration of the left lead (Figure 2), we adopted a single lead mapping approach that was previously described by Minassian et al. [12]. This approach relied on identifying the time of firing of knee extensors (i.e., RF m.) and the timing of mGM m (Figure 5). The stimulation pulses were delivered at 2 Hz and 210 µs and the amplitude of current gradually progressed until a visible contraction was noted in a supine lying position (see Table 2). Once a visible contraction was recognized at a specific amplitude (volts), the frequency of the pulses was delivered at 20, 30, or 40 Hz to ensure the achievement of functional movements. Based on a single lead approach, we tested 10 different configurations of cathodes and anodes (Table 2). As a result, three stimulation configurations (M4, M7, and M8) elicited strong bilateral knee extension when the frequency was adjusted at 20 or 30 Hz. The M10 configuration elicited visible and detectable hip extensor (R > L sides) contractions in the supine lying position (Table 2).

## 2. Targeted Stimulation to Achieve Standing

Based on the findings from Table 1 and Table 2, we limited the selected configurations to M4 and M7 to achieve standing. Each configuration was tested across a range of frequencies (20–30 Hz), pulse durations (210 and 240 µs), and amplitude of current (5.5–6 volts). The selected stimulation parameters were primarily based on findings from Table 2 (see Appendix A). The EMG activity of M4 (−3 and +6) is presented in Figure 6. It is clear that such configurations (20 Hz and 210 µs) targeted the bilateral knee extensors and the right mGM muscle (because of the close proximity to the right lead), with noted firing of the left m GM when the amplitude of the current was adjusted to 5 volts (Figure 6).

Fifteen days post-surgery, the participant failed to do any sit-to stand activity with the scES turned off. With the selected M4 configurations, the participant was capable of maintaining full standing that ranged from 24 s to 150 s (see Appendix A). The longest time was achieved after setting M4 at 20 Hz, pulse durations of 240 µs, and the amplitude of current of 5.5–6 volts. Decreasing the pulse duration at 210 µs resulted in the shortest standing time of 24 s. Setting the frequency at 20 Hz allowed him to enable full knee extension in standing compared to 30 Hz, which resulted in full tetanic contractions of the knee extensors (see Appendix A).

## 3. Targeted Stimulation to Achieve Trunk Control

The participant suffered from impaired trunk control when asked to raise his arms above his head [17]. One of the researchers had to be standing behind him to ensure safety. He failed to maintain a seated balance of more than 10 s (see Appendix A). Stimulation configurations of the targeted rectus abdominus muscle were also recorded in the lying position (Figure 7) at 2 Hz and 240 µs. The stimulation configuration was later used for trunk control training during a seated position. Based on Table 2, M1 and M5 were considered the optimum stimulation configuration for trunk control at 20 Hz. To ensure target stimulation of the trunk muscles, the perturbation of the trunk control was challenged by asking the participant to lift his upper extremities as high as possible above his head (see Appendix A). Using MI at 20 Hz, the participant was able to maintain full trunk control with both arms above his head (see Appendix A).

## 4. Discussion

Here, we present a case of a 25-year-old male with a T3 complete SCI who underwent implantation of bilateral percutaneous surgical leads that were placed originally to cover the distance between T11 and T12. This vertebral region has been previously viewed as the most optimum to cover lumbosacral spinal cord segments from L1–S2 [10]. Lumbosacral spinal cord segments are responsible for inducing either tonic or rhythmic activation of lower extremity muscles in a frequency dependent manner [18]. With scES implantation, this can be translated into standing, stepping, and overground locomotion [3,4,5,12]. Five days after implantation, an X-ray scan confirmed the migration of the left lead outside the epidural space [15].

Despite the migration, the right lead moved caudally at the level of T12, as indicated in Figure 2. Two weeks after implantation, we were capable of configuring the right lead and map the corresponding muscle groups to achieve independent trunk control against upper extremity perturbations and independent standing with a standard walker. This was in agreement with recent findings that indicated that the majority of the patients responded when the cathode resided against the T12 vertebral body after using a straight-line anatomical model of the lumbosacral spinal cord [10].

In the current report, we adopted two mapping strategies that were previously described during applications of scES [12,16]. These two strategies were described by two different research groups: one for paddle implantation and the other for percutaneous implantation. Furthermore, our mapping strategy relied on the activation of two different muscle groups (rectus femoris and gastrocnemius) of distinct segmental distribution at the lumbosacral spinal cord. It is interesting to note that two configuration strategies that resulted in independent standing had −3 or −4 as the cathode. The emphasis of the findings may highlight the significance of locating the cathodes to the corresponding lumbosacral segments to achieve a desired functional movement despite the location of the anodes. This observation was previously emphasized by Rejc, who found that the cathodes in the caudal portion of the paddle were successful in achieving the EMG pattern that led to standing [19].

The use of 2 Hz pulses facilitated the locations of the target muscles by identifying them visually [20]. The findings have great clinical implications for clinicians who are likely to pursue future attempts of percutaneous scES without reliance on EMG. In this report, the visible muscle contraction was later confirmed with a detailed EMG of the target muscles to ensure appropriate activation. We used two muscle groups (RF vs. mGM muscles) with distinct spinal cord segmentation (L2–L4 vs. L5–S1) similar to previous work [10]. Placement of the cathode at T12 previously resulted in the perfect anatomical classification of L2–L3 and L5–S2 spinal cord segments in 34 participants with neurological disorders [10]. This is really important considering the vast variability in the lumbosacral segment length in healthy individuals [21].

In the current work, we chose both inducing (i.e., stimulate specific muscle groups) and enabling supraspinal control with the integration of proprioceptive afferent stimulation of the lower extremity muscles [5]. Target stimulation was previously adopted to ensure appropriate anatomical localization of the specific muscle groups [5]. Once the participant was capable of achieving functional gains (i.e., trunk control and standing), the amplitude of the current was gradually decreased with an increase in the frequency to enable functional restoration of a specific pattern. We have indicated that a frequency set at 20 Hz may yield a tonic activation and provide an opportunity for the participant to enable either supraspinal control or afferent feedback during standing [14,18]. A frequency greater than 20 Hz is also likely to trigger a rhythmic pattern that is important for stepping. When a verbal order was given to lock his knees, the participant successfully managed to perform bilateral knee extension with effective locking at 20 Hz (see Appendix A). A previous review highlighted that a frequency of 21–50 Hz is likely to elicit rhythmic activation of the lumbosacral segment and 30 Hz is considered to be the optimum frequency in humans with SCI [18].

The neurophysiologic mechanisms of immediate motor recovery after two weeks of implantation has recently been highlighted [14]. scES altered the physiological states of the dormant nervous system by increasing the excitably of the spinal networks and supraspinal centers. This can be accomplished by depolarization of the large diameter proprioceptive afferent fibers that results in downstream recruitment of the motor units of the lower extremity muscles [14]. Our participant may have complete injury with spared descending axonal tracts. These dormant descending tracts are likely to be activated with scES, resulting in motor unit recruitment below the level of injury [14]. It is potentially possible that the standing position resulted in load-receptor input, as has been previously suggested [3]. Furthermore, our participant has sensory zones of partial perseveration below T3 without demonstrating any zones of motor preservation. It is still unclear whether these zones of sensory preservation facilitated some extent of volitional motor recovery for trunk control or standing.

Another point of consideration is the midline cathodal approach compared to the lateral approach. It was clear that several groups have focused primarily on the lateral approach to target the passage of the current into afferent fibers of the dorsal neve roots [10]. The midline approach resulted in targeting the entry of the dorsal nerve roots at the junction of the corresponding lumbosacral segment and to trigger inter-neuronal stimulation. Previously, the midline approach has been suggested to accurately target specific spinal cord segments by activating the junctions of the posterior nerve roots [10]. Moving the leads intraoperatively from the midline to lateral locations showed varying responses of EMG activities of the rectus femoris and gastrocnemius muscle. In one of the subjects, midline placement resulted in gastrocnemius activation (27.3%) when the cathode was placed at the L1-vertebral level [10]. For pain management, midline lead placement was primarily intended to target the dorsal column fibers and reduce pain via gait-control mechanism [22]. Other target neural centers have also been suggested to be involved including dorsal horns, sensory root entry zone, and supraspinal centers in pain management [22]. However, the possibility of using a single cylindrical midline that leads to the activation of bilateral segmental muscle contraction has yet to be studied.

Dombovy-Johnson et al. showed that within 20 days of implantation, 88.5% of leads had migrated (86.3% caudal and 2.2% cephalad) in 91 cases [23]. The mean migration distance for leads with caudal migration was only 12.34 ± 12.19 mm based on the antero-posterior radiographs and 16.95 ± 15.68 mm on the lateral radiographs [23]. This low rate of clinically significant migration, which required reoperation, is likely to be attributed to both purposeful cephalad placement and advances in lead programmability. In the current report, we placed the lead contacts higher than the desired location. In addition, placing the IPG between the iliac crest and the 12th rib, ipsilateral to the incision site, ensured that the IPG was in the same anatomical plane as the anchor and entry point, regardless of the body position, thus reducing the lead mobility. Finally, we did have the patient relatively immobile for the first 14 days while the epidural scar forms and lead migration becomes significantly lower. However, the incident occurred as a result of traveling post-operatively to his home in another city. This needs to be avoided in future trials.

In summary, configuration of a single lead scES using a midline approach at the T12–L1 vertebrae resulted in immediate restoration of trunk control and standing in a person with chronic complete SCI, even before the start of any therapeutic training. The targeted stimulation of segmental regions of the lumbosacral segments appeared to enhance the functional recovery via enabling motor control at 20 Hz. The functional gains occurred despite the lead migration, as clearly demonstrated. Future studies are warranted to confirm similar observations in persons with SCI.

## Figures and Tables

**Figure 1 jcm-11-05120-f001:**
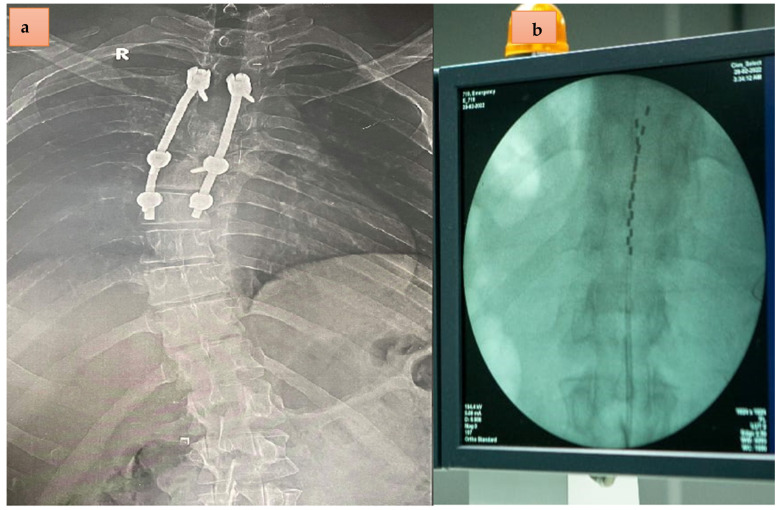
(**a**). The anterior posterior X-ray view of the spine with bilateral rods for spinal fusion and apparent right upper thoracic scoliosis of a T3 complete 25-year-old male with SCI. (**b**). Percutaneous spinal cord epidural stimulation leads placed to cover the distance from T11–T12. The percutaneous leads were placed intentionally above the upper border of T11 to offset for rostral migration, which is likely to happen with percutaneous implantation.

**Figure 2 jcm-11-05120-f002:**
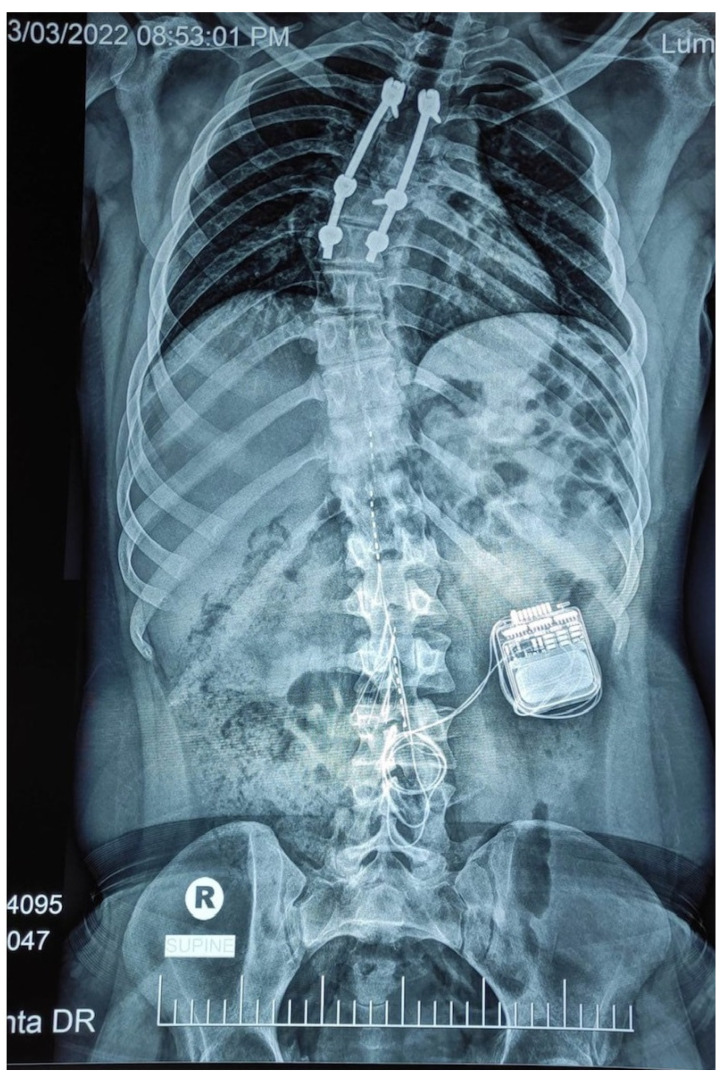
The unexpected left lead migration detected by X-ray. The migration was detected 5 days post-implantation as a result of traveling in a car for close to 250 km to return home. The right lead also migrated to cover the T12–L1 vertebrae.

**Figure 3 jcm-11-05120-f003:**
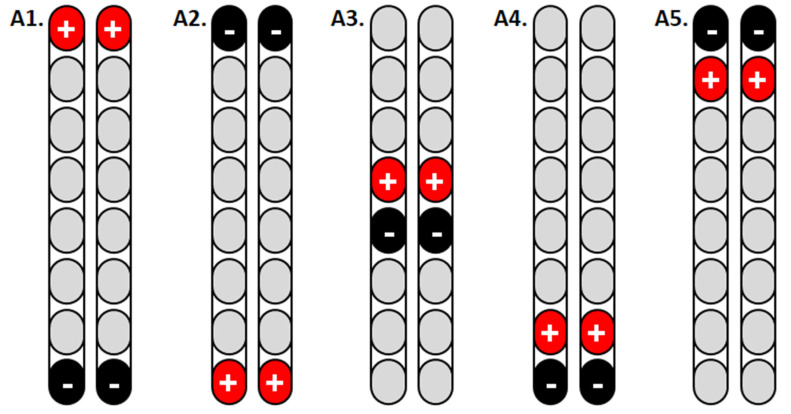
Configuring the percutaneous scES leads with eight contacts each using Sayenko’s approach to target bilateral knee extensor muscles in a lying position. The results of such configurations are listed in Table 1.

**Figure 4 jcm-11-05120-f004:**
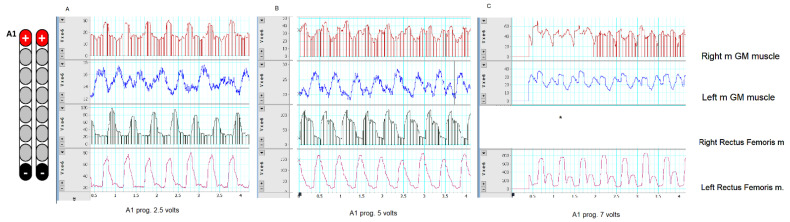
Representative EMG responses of rectus femoris (RF) m. and medial gastrocnemius (mGM) m. evoked by percutaneous scES. The EMG was tested at 2 Hz, 150 µs, and amplitude of current that progressed from 2.5 volts (visible EMG response without visible muscle contractions; (**A**)), 5 volts (visible muscle twitches; (**B**)), and 7.5 volts (strong visible twitches; (**C**)). The EMG data were presented in a filtered format to highlight the distinct responses for both muscle groups. The A1 program was used as a part of Sayenko’s approach before detecting the migration of the left lead. Our results are in agreement with previous findings that noted that the voltage range of 3–8 volts resulted in biased and non-selective stimulation of the RF and mGM muscles. Note the EMG magnitude of rectus femoris m. compared to the medial GM muscle. *, EMG recordings of the right RF muscle was not recorded at 7.5 volts due to a technical error.

**Figure 5 jcm-11-05120-f005:**
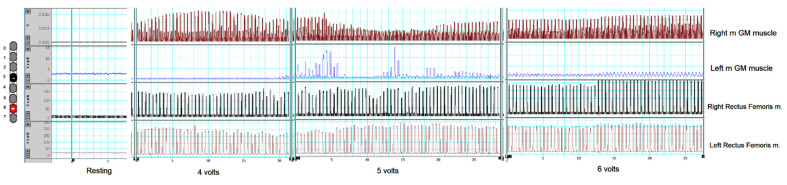
The EMG responses of rectus femoris (RF) m. and medial gastrocnemius (mGM) m. after setting the cathode at −3 and the anode +6. The unilateral configuration of the right lead appeared to achieve and target bilateral firing of the knee extensors and was later on used for full standing using a standard walker.

**Figure 6 jcm-11-05120-f006:**
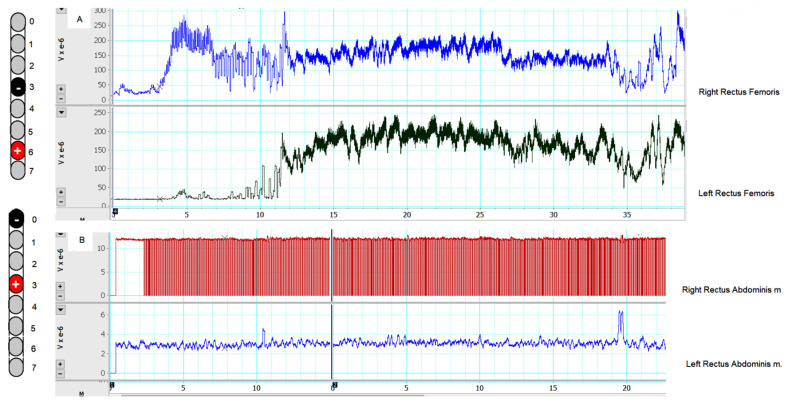
(**A**) The captured EMG activities of the right and left rectus femoris m. during standing activity using a standard walker after configuring the right single lead at −3 and +6 and adjusted at a frequency of 20 Hz, 240 µs, and 5–6 volts. (**B**) The captured EMG activity of the right and left rectus abdominis during sitting at the edge of the mat with use of the upper extremity to perturbate his seated trunk balance after configuring the right lead at −0 and +3 and adjusted at a frequency of 20 Hz, 240 µs, and 5 volts.

**Figure 7 jcm-11-05120-f007:**
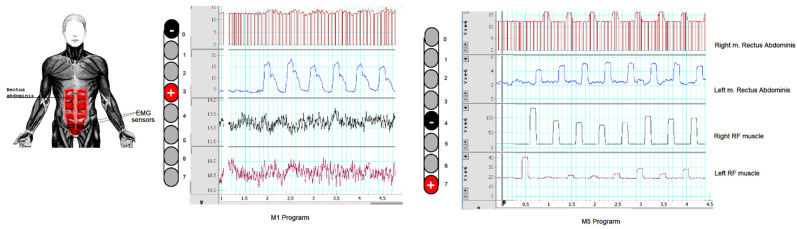
The EMG responses of the right and left rectus abdominis muscle and right and left rectus femoris muscles. A single lead approach was adopted for MI (−0 and +3) and M5 (−4 and +7) with a frequency of 2 Hz, pulse duration of 240 µs, and 6 volts. EMG sensors were placed at the distal 1/3 of the rectus abdominis. It is worth noting that there was no EMG activity detected below 6 volts. M1 resulted in a reasonable isolation of the rectus abdominus muscles from the rectus femoris muscles whereas M5 resulted in bilateral stimulation of the rectus femoris muscles with activation of the rectus abdominis muscles.

**Table 1 jcm-11-05120-t001:** Sayenko’s mapping approach was adopted initially to configure the implanted leads (see Figure 3). After determining the initial responses on day 1 using 2 Hz and 150 µs, the whole entire approach was repeated on the following day (day 2) using 20–30 Hz.

	Day 1 (2 Hz and 150 µs)				Day 2 (20–30 Hz and 210 µs)			
Amplitude (Volts)	2.5	5	7.5	10	2.5	5	7.5	10
A1	No response	Extension with internal rotation (visible contraction)			No response	Left leg flexion was imitated at 3.5 v		
A2	No response	Extension with internal rotation (visible contraction)	Strong bilateral extension		No response	Visible left hip flexion	Participant was capable of enabling left hip flexion at 6.5–7.0 volts	
A3	No response	Emphasis on the right extension	Strong emphasis on the right extension		No response	Bilateral leg extension (potential standing program)	Bilateral leg extension with right side emphasis	Bilateral oscillations of both lower extremities
A4	Right gastrocnemius muscle (visible ankle movement in plantarflexion)	Bilateral emphasis on gastrocnemius and knee extensors	Strong emphasis on gastrocnemius and knee extensors		No response	Visible left hip flexion at 4 volts Bilateral hip flexions at 4.5 volts/		
A5	No response	No response	No response	Left side knee extensors and gluteus maximums muscle	Program was excluded based on the results of day 1			

**Table 2 jcm-11-05120-t002:** We used two distinct muscle group segmental approaches to identify the optimum stimulation configurations (cathodes and anodes) that led to the activation of bilateral RF followed by GM muscle groups using a midline single lead. All pulses were delivered at 2 Hz and 210 µs.

				Amplitude (Volts)							
Program	Cathode	Anode	Muscles	1	2	3	4	5	6	7	Abdominal Muscle Stimulation (Y/N)
M1	0	3	Left RF on					x	x x	x x	Y
			Right RF on								
			Left GM on								
			Right GM on								
M2	1	4	Left RF on				x	x x			N
			Right RF on								
			Left GM on								
			Right GM on								
M3	2	5	Left RF on				x x	x x			N
			Right RF on								
			Left GM on								
			Right GM on								
M4	3	6	Left RF on			x x	x x				N
			Right RF on								
			Left GM on								
			Right GM on								
M5	4	7	Left RF on			x x	x x x x	x x x x	x x x x		Y
			Right RF on								
			Left GM on								
			Right GM on								
M6	3	0	Left RF on			x	x x	x x			N
			Right RF on								
			Left GM on								
			Right GM on								
M7	4	3	Left RF on			x	x x				N
			Right RF on								
			Left GM on								
			Right GM on								
M8	5	2	Left RF on			x	x x				N
			Right RF on								
			Left GM on								
			Right GM on								
M9	6	3	Left RF on			x x	x x				N
			Right RF on								
			Left GM on								
			Right GM on								
M10	7	4	Left RF on		x x	x x x x	x x x x				N
			Right RF on								
			Left GM on								
			Right GM on								

Each configuration was then tested at 20–40 Hz to determine the optimum functional movements that can be achieved with specific targeted stimulation.

## Data Availability

The data presented in this study are available on request from the corresponding author. The data are not publicly available due to potential patient privacy risks.

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
