# Peer review of "Single Lead Epidural Spinal Cord Stimulation Targeted Trunk Control and Standing in Complete Paraplegia"

_jcm, 2022, doi:10.3390/jcm11175120_

Round 1
Reviewer 1 Report
The theme of the article is relevant and suitable for publication in this journal. However, the article needs improvements:
1) The authors should show MRI, X-ray data pre- and post-surgery. In addition, need to describe the neurological examination data (deep tendon reflex, Babinski reflex, sensory disturbance area, touch/pinprick and position sense etc.) with change over time.
2) If the authors revealed the temporal neurological recovery following electrode stimulation scientifically, you need the electrophysiological data, for example, EMG data in lower limbs muscle before and after stimulation because the authors mentioned about standing.
3) In discussion, the author needs to mention more about the mechanism of motor recovery.
4) This abstract information is not enough, we could not well understand the contents of this article. Please rewrite the abstract to be easily realize the meaning of your paper for readers.
Author Response
The theme of the article is relevant and suitable for publication in this journal. However, the article needs improvements:
- The authors should show MRI, X-ray data pre- and post-surgery. In addition, need to describe the neurological examination data (deep tendon reflex, Babinski reflex, sensory disturbance area, touch/pinprick and position sense etc.) with change over time.
Answer line 67-70: We have presented x-ray scans on the patient pre-(figure 1a) and fluoroscopy image (figure 1b). We felt that this is clinically sufficient so as not to expose our patient to unnecessary dose of radiations. We totally agree with the reviewer, and we have presented a full AIS exam that was conducted before the surgery. The data presented in the current study occurred within 3 weeks from the surgery, so we did not expect any changes in AIS classification to occur in this short period and we did not perform AIS during this period (Suppl. Figure 1).
- If the authors revealed the temporal neurological recovery following electrode stimulation scientifically, you need the electrophysiological data, for example, EMG data in lower limbs muscle before and after stimulation because the authors mentioned about standing.
Answer: We are really surprised by the comment of the reviewer and apologize if you did not have access to figures 4 and 5, because we the have the EMG data of the lower extremity muscles listed in Figure 4, Figure 5 and 7. The way we have our EMG data presented with the Epidural stimulation off (i.e. resting) and with the epidural stimulation on. We do not have any EMG data before the implantation
- In discussion, the author needs to mention more about the mechanism of motor recovery.
Answer line 243-249: Thank you so much for your comment, we have recently published the following article (reference #12) and we have used as a reference to discuss the mechanism of motor recovery in persons with SCI. We have added an entire paragraph to address your concern.
- This abstract information is not enough, we could not well understand the contents of this article. Please rewrite the abstract to be easily realize the meaning of your paper for readers.
Answer: We have attempted to clarify some aspects of the abstract to improve readability.

Reviewer 2 Report
This case report brings some interesting insights on the possible usefulness of Spinal Cord Stimulation (SCS) in the rehabilitation of Spinal Cord Injury patients.
In literature, spinal cord stimulation, obtained through the implant with an epidural lead, is mainly abbreviated in SCS, so I suggest this acronym.
The patient described was evaluated as a AIS A because of the absence of sacral sensory and motor function but, as the Authors state that, after SCS, “the patient successfully managed to perform bilateral knee extension” when asked, and it is hypothesized that a supraspinal control was involved, we should imagine a partial functional preservation of the motor and sensory pathways below the level of the lesion; the presence of Zones of Partial Preservation (ZPP) should be described in this case because they are essential to understand and to foresee the possibility of some voluntary movement recovery.
The Authors focused on the persistence of functional improvement with SCS despite the migration of one lead out of the epidural space and the caudal migration of the second lead, but the more interesting data is the capability of a single midline lead stimulation at T12-L1 to activate motor contraction in some muscle groups useful to improve rehabilitation.
It is not described if an intra-operative trial was performed to obtain some muscle contraction and eventually EMG recording.
The midline lead placement is intended to target mainly dorsal column fibers in order to obtain pain management through gate control (according to older theories) but other targets have been hypothesized: dorsal horn, sensory roots entry zone, supraspinal centers. The possibility to activate bilateral segmental muscle contraction with a single cylindrical midline lead has not been yet studied to my knowledge.
Jensen MP, Brownstone RM. Mechanisms of spinal cord stimulation for the treatment of pain: Still in the dark after 50 years. EJP. 2019; 23: 652,659.
Caylor J et al. Spinal cord stimulation in chronic pain: evidence and theory for mechanisms of action. Bioelectron Med 5, 12 (2019). https://doi.org/10.1186/s42234-019
Author Response
This case report brings some interesting insights on the possible usefulness of Spinal Cord Stimulation (SCS) in the rehabilitation of Spinal Cord Injury patients.
Answer: We would like to thank the reviewer for his/her interest in our work.
In literature, spinal cord stimulation, obtained through the implant with an epidural lead, is mainly abbreviated in SCS, so I suggest this acronym.
Answer: We would like thank the reviewer. We have previously used this acronym in our previous publications. For consistency purpose, we would like to use the same acronym scES.
- Gorgey AS, Gill S, Holman ME, Davis JC, Atri R, Bai O, Goetz L, Lester DL, Trainer R, Lavis TD. The feasibility of using exoskeletal-assisted walking with epidural stimulation: a case report study. Ann Clin Transl Neurol. 2020 Feb;7(2):259-265.
- Gorgey AS, Sutor TW, Goldsmith JA, Ennasr AN, Lavis TD, Cifu DX, Trainer R. Epidural stimulation with locomotor training ameliorates unstable blood pressure after tetraplegia. A case report. Ann Clin Transl Neurol. 2022 Feb;9(2):232-238.
The patient described was evaluated as a AIS A because of the absence of sacral sensory and motor function but, as the Authors state that, after SCS, “the patient successfully managed to perform bilateral knee extension” when asked, and it is hypothesized that a supraspinal control was involved, we should imagine a partial functional preservation of the motor and sensory pathways below the level of the lesion; the presence of Zones of Partial Preservation (ZPP) should be described in this case because they are essential to understand and to foresee the possibility of some voluntary movement recovery.
Answer line 251-254: We would like to thank the reviewer for his feedback, we are now provided a detailed AIS classification that clearly showed that the patient has some zones of sensory partial preservation below T3 without any zones of motor preservation below the level of injury. We have also addressed your point in the discussion section.
The Authors focused on the persistence of functional improvement with SCS despite the migration of one lead out of the epidural space and the caudal migration of the second lead, but the more interesting data is the capability of a single midline lead stimulation at T12-L1 to activate motor contraction in some muscle groups useful to improve rehabilitation.
Answer line 279: We have definitely considered your great point and we have updated both the abstracts and the conclusion of the study to reflect your point.
It is not described if an intra-operative trial was performed to obtain some muscle contraction and eventually EMG recording.
Answer line 87-88: we have performed intra-operative trial to ensure appropriate muscle contraction without any EMG recordings. It is difficult to access the surgery room with EMG sensors in this setting.
The midline lead placement is intended to target mainly dorsal column fibers in order to obtain pain management through gate control (according to older theories) but other targets have been hypothesized: dorsal horn, sensory roots entry zone, supraspinal centers. The possibility to activate bilateral segmental muscle contraction with a single cylindrical midline lead has not been yet studied to my knowledge.
Jensen MP, Brownstone RM. Mechanisms of spinal cord stimulation for the treatment of pain: Still in the dark after 50 years. EJP. 2019; 23: 652,659.
Caylor J et al. Spinal cord stimulation in chronic pain: evidence and theory for mechanisms of action. Bioelectron Med 5, 12 (2019). https://doi.org/10.1186/s42234-019
Answer line 267-270: Thank you so much, your point is well taken and we have discussed it and referenced one of the recommended articles.

Round 2
Reviewer 1 Report
Thank you very much for your big effort in improving the paper following my notes.
Author Response
Dear Reviewer,
Thank you so much for your excellent feedback and comments. We truly appreciate your effort reviewing our work.
Reviewer 2 Report
The Authors improved the content of the paper but the editing still needs some review.
In line 41 I would change in "mapping of the lumbosacral segments of the spinal cord"
Lines 47-48: remove locomotor and restored
Line 66: muscles
Line 68: 3 years and 8 months before
Line 78; what was done only for research purposes? the consent or the implant?
Lines 81-82: the migration of the leads could be possibly attributed to their lenght (?)
Line 86: the lead is not in L1
Figure 2: in the image it is possible to see 4 contacts at the level of T11: where do they come from?
Line 140: you use the present "is" but all other verbal times are in the past form
Libne 145: Palatable should be changed with a more appropriate adjactive
Line 183: less should be changed with more or change the verb
Line 190: head
Line 193: single lead
Line 202: remove "more"
Line 208: it would be better "moved caudally at the level of T12"
Line 224: was instead of has
Line 247: ypu should comment on the fact that the a patient without any motor activity sparing good moove voluntarily
Line 289: incident
Author Response
Reviewer 2
We would like to thank the reviewer for his/her time and effort reviewing and providing insightful comments to our manuscript. We believe that your comments greatly enriched the quality of the work.
The Authors improved the content of the paper, but the editing still needs some review.
We attempted to follow your directions and make corrections based on your feedback.
- In line 41 I would change in "mapping of the lumbosacral segments of the spinal cord"
Answer: Changes have been made as suggested.
- Lines 47-48: remove locomotor and restored
Thank you
- Line 66: muscles
Thank you.
- Line 68: 3 years and 8 months before
Thank you, this has been fixed.
- Line 78; what was done only for research purposes? the consent or the implant?
Thank you. The procedure of scEC implantation was done for research purpose. The phrase scES implantation was added to clarify the meaning.
- Lines 81-82: the migration of the leads could be possibly attributed to their length (?)
We believe that due to longer length of the leads (~90 cm), it may possibly contribute to migration.
- Line 86: the lead is not in L1
Line 83: we totally agree with the reviewer and this has changed to T12.
- Figure 2: in the image it is possible to see 4 contacts at the level of T11: where do they come from?
This is an excellent point, this is actually an anatomical print of the left migrated lead and they can be noticeable in some x-ray scans.
- Line 140: you use the present "is" but all other verbal times are in the past form
Line 135: Thank you, this was fixed.
- Line 145: Palatable should be changed with a more appropriate adjective
Line 141: Thank you, this has changed to detectable.
- Line 183: less should be changed with more or change the verb
Line 178: Thank you, this was corrected to more than
- Line 190: head
Line 184: Thank you, this was corrected.
- Line 193: single lead
Line 188: Thank you, this was corrected.
- Line 202: remove "more"
Thank you
Line 208: it would be better "moved caudally at the level of T12"
Line 203: completed, thank you.
Line 224: was instead of has
Line 219: Done
Line 247: you should comment on the fact that the a patient without any motor activity sparing good move voluntarily
Thank you so much for your comment. We have added the following two sentences. “Our participant may have a discomplete injury with spared descending axonal tracts. These dormant descending tracts are likely to be activated with scES resulting in motor unit recruitment below the level of injury (14).
Line 289: incident
Line 283: Done
